# Does Malpositioning of Pedicle Screws Affect Biomechanical Stability in a Novel Quasistatic Test Setup?

**DOI:** 10.3390/bioengineering12070781

**Published:** 2025-07-18

**Authors:** Stefan Schleifenbaum, Florian Metzner, Janine Schultze, Sascha Kurz, Christoph-Eckhard Heyde, Philipp Pieroh

**Affiliations:** 1ZESBO—Center for Research on Musculoskeletal Systems, Semmelweisstraße 14, 04103 Leipzig, Germany; florian.metzner@medizin.uni-leipzig.de (F.M.); sascha.kurz@medizin.uni-leipzig.de (S.K.);; 2Department of Orthopedic, Trauma, and Plastic Surgery, University of Leipzig, Liebigstraße 20, 04103 Leipzig, Germany; philipp.pieroh@medizin.uni-leipzig.de

**Keywords:** pedicle screw, malposition, thoracolumbar, biomechanics, stability, screw loosening

## Abstract

Pedicle screw fixation is a common spinal surgery technique, but concerns remain about stability when screws are malpositioned. Traditional in vitro pull-out tests assess anchorage but lack physiological accuracy. This study examined the stability of correctly placed and intentionally malpositioned pedicle screws on forty vertebrae from five cadavers. Optimal screw paths were planned via CT scans and applied using 3D-printed guides. Four malposition types—medial, lateral, superior, and superior-lateral—were created by shifting the original trajectory. A custom setup applied three consecutive cycles of tensile and compressive load from 50 N to 200 N. Screw inclination under load was measured with a 3D optical system. The results showed increasing screw inclination with higher forces, reaching about 1° at 50 N and 2° at 100 N, similar in both load directions. Significant differences in inclination were only found at 100 N tensile load, where malpositioned screws showed a lower inclination. Overall, malpositioning had no major effect on screw loosening. These findings suggest that minor deviations in screw placement do not significantly compromise mechanical stability. Clinically, the main concern with malpositioning lies in the potential for injury to nearby structures rather than reduced screw fixation strength.

## 1. Introduction

Appropriate pedicle screw placement is essential for the stability of posterior fixation constructs of the spine [1,2]. Laterally, medially, or cranially misplaced screws might impair the implant’s biomechanical properties, affecting the stability and long-term safety of the treatment [3,4]. Gertzbein and Robbins [5] provided a systematic classification of malpositioning based on the extent to which the screw breaches the pedicle cortex, focusing on possible neurological deficits [3,5,6]. In particular, laterally or medially misplaced screws might become anchor increasing pull-out strength and reducding the risk of implant failure [4,7]. For example, it has been shown that medial and lateral malpositioning significantly reduce the tensile and torsional strength of the screws, increasing the risk of implant failure [4].

Recent donor-based biomechanical studies have shown that stability differs between medial and lateral malpositioning. Studies have demonstrated a general reduction in screw load-bearing capacity, and the reduction was less pronounced in medial than in lateral malpositioning [7,8,9]. Furthermore, the repositioning of the screws after initial malpositioning (e.g., using the previous pilot hole) did not significantly reduce the fixation strength [10,11]). The pull-out strength after the reinsertion of the screws due to lateral or endplate-related breakthrough is also an important parameter that underlines the clinical relevance of the screw position. It has been shown that the pull-out force has not changed and therefore the screw can be reinserted [10,11,12].

This study investigated the biomechanical stability of correctly placed and misplaced pedicle screws. In contrast to uniaxial pull-out tests [7,8,9], we analyzed the cycling cranio-caudal stability by means of a biomechanical test setup presented by Javers et al. [13]. This should be performed by measuring the tilt angle between the faulty and optimum screw. Can the comparison of different malpositionings with the optimal screw position provide insights that emphasize the importance of precise placement and provide information for clinical practice?

## 2. Materials and Methods

### 2.1. Ethical Statement

All body donors gave their informed and written consent to the donation of their bodies for teaching and research purposes while alive. Being part of the body donor program regulated by the Saxonian Death and Funeral Act of 1994 (third section, paragraph 18 item 8), institutional approval for the use of the post-mortem tissues of human body donors was obtained from the Institute of Anatomy (University of Leipzig) by the Ethics Committee of the University of Leipzig Medical Center (ethical approval No. 129/21-ck). The authors declare that all experiments were conducted according to the principles of the Declaration of Helsinki (as revised in 2013).

### 2.2. Specimen and Tissue Preparation

Ten vertebrae (T8-L5) from five body donors (mean age 77.2 ± 13.2 years; n = 4 male, n = 1 female) with different bone densities (each one non-osteoporotic and osteopenic, n = 3 osteoporosis) were taken fresh frozen. The specimens were stored at −80 °C and thawed to room temperature (20 °C) 24 h before testing. On testing day, all vertebrae were separated, and all soft tissue was removed. Osteoporosis was classified by determining the volumetric bone mineral content (vBMD) using quantitative CT scans in which a calcium hydroxyapatite phantom was inserted. The mean Hounsfield unit (HU) of each vertebral cancellous bone was determined using Mimics Innovation Suite 24 (Materialise, Leuven, Belgium) [14], the corresponding vBMD was calculated by linear regression [15] for each lumbar vertebra, and a mean value for each donor was calculated.

The samples were then embedded in aluminum sleeves using rapid casting resin consisting of the three components, RenCast FC 52/53 isocyanate, FC 53 polyol (Huntsman Advanced Materials, Basel, Switzerland) and aluminum hydroxide (filler DT 082, Gößl + Pfaff GmbH, Brautlach/Karlskron, Germany), at a ratio of 1:1:3 equivalent as published previously [13,16].

### 2.3. Planning and Screw Positioning

The traditional transpedicular trajectory was defined as the correct screw position (c) for the experimental design. They were individually planned using CAD software (Rhinoceros Version 7 SR37, Robert McNeel & Associates, Seattle, WA, USA), taking all spatial directions into account. Starting from the correct trajectory, four malpositionings were defined as follows: medial (m), lateral (l), superior (s), and superior-lateral (sl). Malpositioning was generated by defined parallel shifting from the correct trajectory (Figure 1) without angular correction. The displacement value corresponds to the distance by which the screw breaks out of the pedicle wall and is based on previous reports [5] (Figure 1), All malpositioning is classified as Grade C, following the classification of Gertzbein and Robbins [5].

The screw trajectories were distributed evenly across the donors and randomly across the pedicles (left/right) to avoid possible side dependencies (Table 1).

Individual drilling templates for the respective vertebrae were then designed using the above-mentioned CAD software and 3D printed using the Polyjet printing process (Stratasys J850 DAP, Stratasys Inc., Minnetonka, MN, USA) from VeroPureWhite material. After determining the entry points, the traditional trajectory along the pedicle was defined (see Figure 2A), and the appropriate screw diameter was selected, which is represented as a cylinder in the software (see Figure 2B). This was followed by the aforementioned parallel displacement of the correct screw position in the desired direction, using the predefined offset (1). Connecting elements and supports were then designed onto the drilling templates to ensure secure placement on the vertebra (Figure 2C). The templates were printed.

M.U.S.T. pedicle screws from Medacta International (M.U.S.T., Medacta International, Castel San Pietro, Switzerland) were utilized in the experimental setting, along with the corresponding original instruments for inserting and fixing the screw–rod system in the test bench. The fixation of the rod was executed in accordance with the manufacturer’s specifications.

### 2.4. Biomechanical Experiments

An optimized test setup in accordance with ASTM F1717-15 [17] was used for the biomechanical tests, similar to the dynamic tests used by Javers et al. and Schleifenbaum et al. [13,16]. The setup is designed to enable standard positioning of the pedicle screw relative to the lower axis of rotation (Figure 3). Testing was then performed in a servo-electric testing machine (RetroLine Z10, Zwick/Roell GmbH & Co. KG, Ulm, Germany) equipped with a 2.5 kN force sensor. The movement of the screw head relative to the bone was recorded during the test using an optical 3D measurement system (ZEISS ARAMIS 3D Camera, Carl Zeiss GOM Metrology GmbH, Braunschweig, Germany). This measurement can output the movement of the screw in degrees and thus describes the movement of the screw that can lead to cut-outs from the vertebra, among other things.

The load application was executed as a quasi-static test consisting of compressive and tensile loads in the craniocaudal direction. These loads were applied with 5 N/s to the pedicle screw via the implant rod in three cycles. The target force values increased gradually, reaching ±50 N (first cycle), ±100 N (second cycle), and ±200 N (third cycle) (Figure 3). In the event of significant loosening of the screws during compression loading, a risk of collision was identified due to the compact design of the test setup. This was prevented by premature manual shutdown. The following exclusion criteria were defined for data evaluation:Test not possible (no pedicle screw instrumented due to fracture, for example);Embedding error (e.g., embedded too deeply, screw in embedding compound);Error during instrumentation (e.g., drill guide slipped, screw position out of plane)Incorrect data recording;Side comparison not possible (because data evaluation of the comparison pedicle was excluded due to the above criteria).

### 2.5. Statistical Analysis

The data were processed using MS Excel (MS Office 2016, Microsoft Corporation, Redmond, WA, USA) and statistically evaluated using SPSS software (IBM SPSS Statistics 24.0, IBM Corporation, Armonk, NY, USA). The study investigated whether the pedicle screw inclination depends on correct or incorrect screw positioning. A paired Wilcoxon test was used to account for the differences between misplaced and correctly implanted screws, after a test of normal distribution (Kolmogorov–Smirnov-Test). The significance level α was set to 0.05.

## 3. Results

During the preparation and execution of the mechanical tests, a total of 17 vertebrae were excluded from the 40 vertebrae planned, with a total of 80 screws. The previously defined exclusion criteria were applied. The excluded screws were distributed as follows: six times criterion 1, eight times criterion 2, five times criterion 3, and seven times criterion 4. For a comparative evaluation, vertebrae with usable data recording were required that were instrumented as planned on both sides (criterion 5). From this, twentythree vertebrae with 46 screws were used for the final evaluation. Twenty-three vertebrae instrumented with a misplaced pedicle screw on one side could be used for evaluation, nine of which had medial (m), six lateral (l), three superior (s), and five superolateral (sl) malpositionings, as shown in Figure 4. Due to the low number of individual misalignments (l, m, s, sl), these were grouped together as malpositioning (F).

A total of eight pedicle screws completed the load test: four with correct positioning, two with lateral malpositioning, and two with medial malpositioning. The remaining tests resulted in screw loosening or severe tilting during the third cycle of the test. For this reason, only data from loads between ±50 N and ±100 N of all 46 screws were evaluated.

### Comparison of Malpositioning

The Wilcoxon test showed that there were only significant differences between correctly and incorrectly positioned screws at a tensile load of 100 N (Table 2). The said difference in inclination at 100 N is approximately 2° (Figure 5). The box plots indicate increasing dispersion of the inclination with increasing loads (Figure 5), but no differences between the groups are apparent.

## 4. Discussion

The study shows no biomechanical difference between the incorrectly placed screws and the correctly placed screws. There is a slight tendency for the misplaced screws to have a slightly lower tilt. This is supported by the significant differences at 100 N and the visualization of the box plots. However, the high data variability reduces the statistical power, limiting the interpretability of these trends.

From a biomechanical perspective, misplaced screws may remain in the vertebral body, provided there are no clinical concerns or symptoms. Previous studies have shown that even suboptimal placed screws can provide sufficient primary stability, unless there is a perforation of cortical structures or a risk to neurovascular elements [3,6,9,18]. This is further supported by pull-out tests, which assess the maximum extraction force of screws and, in some cases, show no significant difference between correctly and slightly misplaced screws [7,9].

Most recently, a biomechanical study revealed a higher pull-out force for medially misplaced lumbar pedicle screws compared to correctly implanted screws, whereas laterally misplaced showed a reduced pull-out force [4]. Therefore, medially misplaced screw should be reinserted to prevent neurological damage, whereas laterally misplaced screws should be reinserted to improve construct stability [4]. Clinically, in a 12-month follow-up, lateral breaches were found to have no impact on functional outcome and fusion but affected life quality and the postoperative improvement of leg and back pain [19].

However, the transferability of pedicle screw pull-out to clinical scenarios is limited. As demonstrated by Jarvers et al. [13], the results from biomechanical test environments can only be applied to in vivo conditions to a limited extent. A key advantage of the test setup used in this study lies in its time efficiency while still capturing dynamic loading scenarios. In contrast to traditional pull-out tests, which only simulate static maximum loads, the test method employed here allows for the simulation of physiologically relevant cyclic loads. This corresponds more closely to the actual mechanical load on the spine in everyday life [20]. Song et al. [21] showed in an FE study that the cranial–caudal movement corresponds to failure more than the pull-out, This supports the thesis of our test-setup as a practical addition to established test procedures and expands the biomechanical assessment of screw stability under quasi in vivo conditions.

Historically, a malpositioning of >4 mm was defined as a risk for neurovascular injury [5]. But there is low agreement between surgeons on which screws should be revised. A survey revealed that in asymptomatic patients, only 25% of surgeons tended to remove medially or inferiorly misplaced screws with >6 mm breach [22]: neurological symptoms were a strong indication for screw correction [22].

The rate of cortical breach or malpositioning of pedicle screws ranged from 2 to 50% depending on the definition [23]. Screw malpositioning accounts for the top four reasons leading to reoperation during the first 30 days following spine surgery [24]. In pediatric spine deformity surgery, screw malpositioning was the most frequently reported complication and reason for reoperation [25].

## 5. Limitations

A limitation of this study is that the investigations were conducted exclusively on human donor specimens, which significantly restricted the availability and number of samples. Due to the limited sample size, the statistical power of the results is constrained, potentially affecting the generalizability of the findings. To ensure a broad coverage of different bone density qualities, the specimens were carefully selected based on their bone quality and filtered through strict exclusion criteria. This approach helped to ensure data quality and improve comparability of the results. Another aspect is that, due to the limited number of samples, a paired comparison was not always possible. Consequently, some specimens had to be excluded from the analysis. This limitation is common in biomechanical studies using human specimens, as the availability of suitable samples poses a significant challenge [26].

## 6. Conclusions

Our results show that pedicle screw malpositioning of this extent has no significant influence on screw loosening. The biomechanical test results show that the malpositioning of pedicle screws does not necessarily lead to reduced stability of the implant in the bone. From a clinical point of view, malpositionings seem to have a greater impact on the potential injury to surrounding structures. The test procedure also reflects the pull-out results but is more physiological in terms of load application and therefore more transferable to clinical contexts in terms of perspective.

## Figures and Tables

**Figure 1 bioengineering-12-00781-f001:**
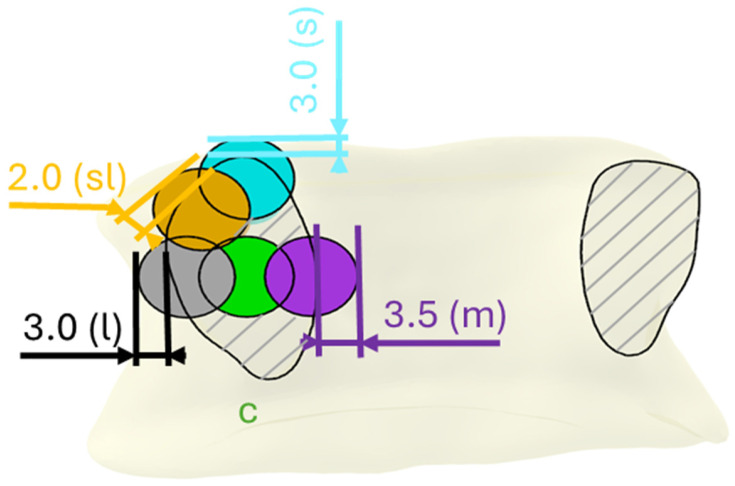
Planned screw trajectories within the pedicle: transpedicular coronal cross-section. Correct (c), lateral (l), superior-lateral (sl), superior (s), and medial (m) screw trajectories with indicated amount of cortical breach (mm).

**Figure 2 bioengineering-12-00781-f002:**
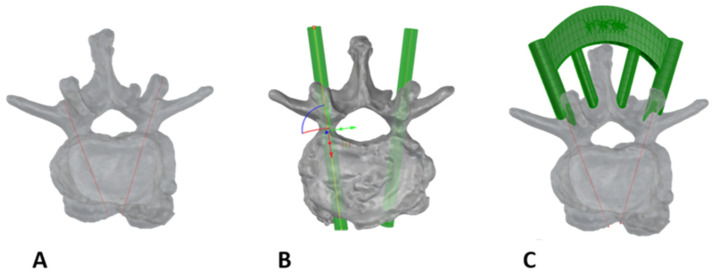
The procedure for designing drilling templates. First, the trajectory was set as a curve (**A**), followed by creating cylinders as a preview of the screw diameter (**B**). Finally, a predefined design was applied to the curves, and the vertebral geometry was subtracted (**C**).

**Figure 3 bioengineering-12-00781-f003:**
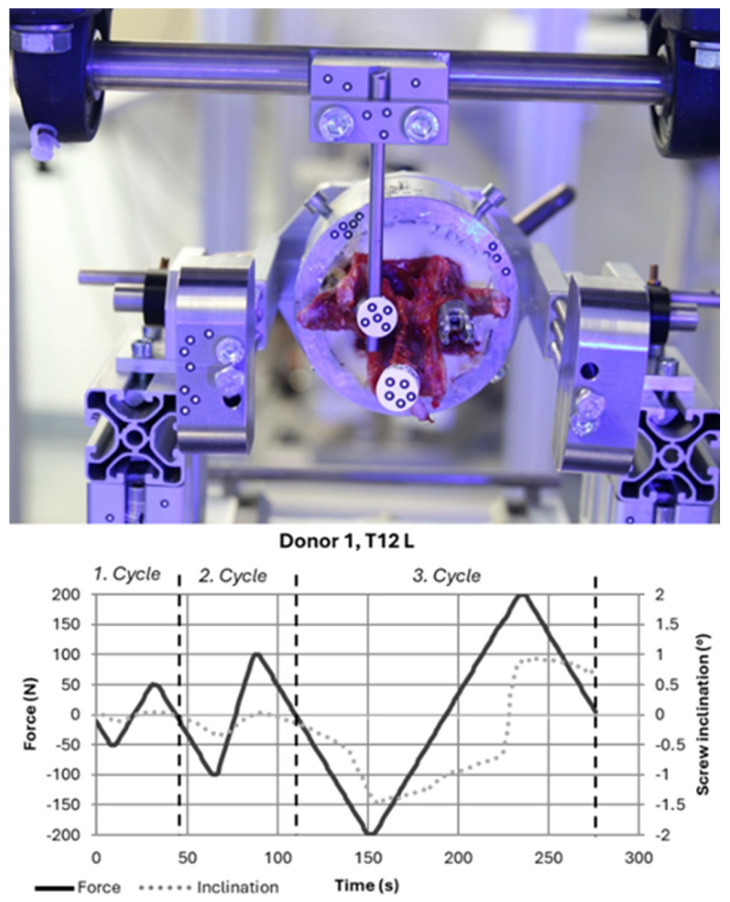
A mounted specimen with optical markers on the screw head and the spinal process (**top**). Axial force and screw inclination as a function of time (**bottom**).

**Figure 4 bioengineering-12-00781-f004:**
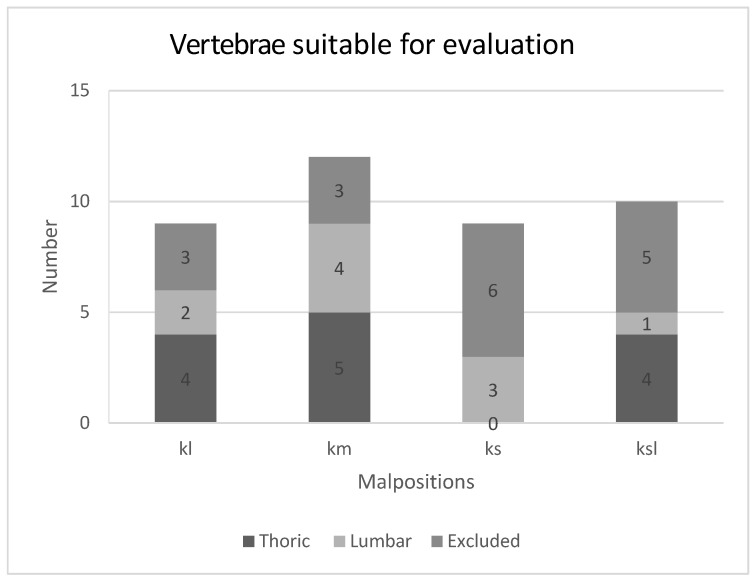
Distribution of suitable vertebrae for data evaluation grouped into each malpositioning (lateral—l; medial—m; superior—s; superior-lateral—sl).

**Figure 5 bioengineering-12-00781-f005:**
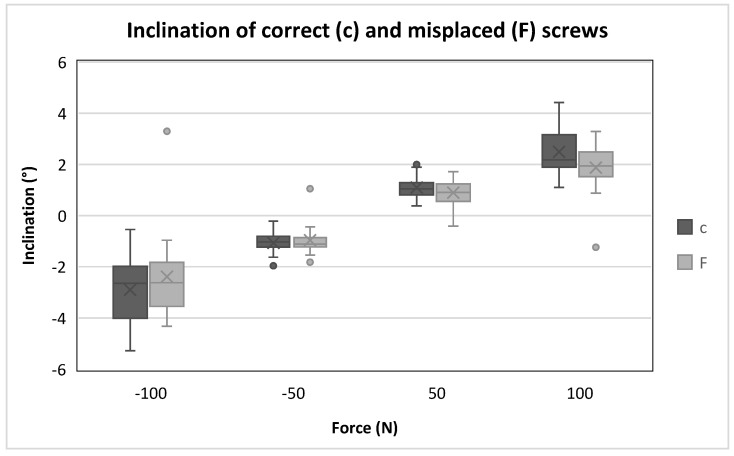
Boxplots showing the inclination of correct and misplaced pedicle screws at each maximum or minimum force.

**Table 1 bioengineering-12-00781-t001:** Each trajectory (medial—m; lateral—l; superior—s; superior-lateral—sl; correct—c) occurred twice per donor, and the side selection (left or right) was randomized.

Donor	1		2		3		4		5	
Level	Left	Right	Left	Right	Left	Right	Left	Right	Left	Right
T8	m	c	c	sl	s	c	c	c	l	c
T9	l	c	m	c	c	sl	c	s	c	c
T10	c	c	c	l	m	c	c	sl	s	c
T11	c	s	c	c	c	l	c	m	sl	c
T12	c	sl	s	c	c	c	c	l	c	m
L1	c	m	c	sl	s	c	c	c	l	c
L2	l	c	m	c	c	sl	c	s	c	c
L3	c	c	l	c	c	m	sl	c	c	s
L4	c	s	c	c	l	c	m	c	c	sl
L5	sl	c	c	s	c	c	c	l	c	m

**Table 2 bioengineering-12-00781-t002:** Comparison of median inclination at maximum and minimum forces during the first two cycles for correctly and misplaced screws using the Wilcoxon test.

Cycle	Force (N)	Correct Inclination (°)	Misplacement Inclination (°)	*p*-Value
1	−50	−1.0	−1.1	0.615
	50	1.0	0.9	0.277
2	−100	−2.7	−2.6	0.355
	100	2.2	1.9	0.026

## Data Availability

The data that support the findings of this study are available on reasonable request from the corresponding author.

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
