# Peer review of "Does Malpositioning of Pedicle Screws Affect Biomechanical Stability in a Novel Quasistatic Test Setup?"

_bioengineering, 2025, doi:10.3390/bioengineering12070781_

Round 1
Reviewer 1 Report
Comments and Suggestions for Authors
There are numerous assumptions that inhibit the reliability of the findings and the clinical applicability.
Firstly the position of the screws were assessed in the axial plane based on the orientation and location in the pedicle. The pedicle itself is not the sole source of fixation of a pedicle screw, and distal characteristics play a role. Whether the distal pedicle screw is entirely in the vertebral body, or takes an outside in trajectory can have an influence.
Was bone density assumed to be the same at all levels, or was this actually tested per level.
Was the fill of the pedicle in relation to the screw diameter assessed.
Was the sagittal plane assessed.
The soft tissue assumptions are not clearly stated and should be elaborated upon.
A bit concern is whether the occurrence of a fusion and the likelihood to progress to a fusion was assessed. This plays a major influence on not only the long term stability of the construct, but the likelihood to develop symptoms and require a revision. This is glaringly absent from the analysis.
What direction were the forces repeated to pull out tested. Were only uniplanar considerations made. Were coupled motions evaluated?
Author Response
There are numerous assumptions that inhibit the reliability of the findings and the clinical applicability.
Comments
Firstly the position of the screws were assessed in the axial plane based on the orientation and location in the pedicle. The pedicle itself is not the sole source of fixation of a pedicle screw, and distal characteristics play a role. Whether the distal pedicle screw is entirely in the vertebral body, or takes an outside in trajectory can have an influence.
Response: All screws were planned in standard trajectory within 3D-CAD software. Thus, all oriantations were included during planning. As whe only tested the vetrebrae itself, without ribs and removed from all soft tissues, all screw were planned in a straightforward transpedicular trajectory. Specifications were included in Lines 87-93.
Comments
Was bone density assumed to be the same at all levels, or was this actually tested per level.
Response: Bone density was assesed for each vertebra and an average value was calculated in order to classify each donor regarding their bone quality. See Lines 77-80
Comments
Was the fill of the pedicle in relation to the screw diameter assessed.
Response: No we didn’t asses this issue. Due to our study design which compared correct and malpositioned screws, it seemed not necessary to gather this information because al malpositioned screws are placed out of the cortex.
Comments
Was the sagittal plane assessed.
Response: Each screw was planned parallel between the endplates. Within our CAD-Software we placed a midvertebral plane for each vertebrae. This plane was used as a starting point for screw planning and further positioning was done using 3D-View. Said starting plane was also needed as reference for creating the drilling guides. So yes, the sagital plane was addressed. See Lines 87-93.
Comments
The soft tissue assumptions are not clearly stated and should be elaborated upon.
Response: All soft tissues including the intervertebral disks and surrounding soft tissues were removed prior to embedding. See Lines 74-75
Comments
A bit concern is whether the occurrence of a fusion and the likelihood to progress to a fusion was assessed. This plays a major influence on not only the long term stability of the construct, but the likelihood to develop symptoms and require a revision. This is glaringly absent from the analysis.
Response The aim of this study was to investigate the bone-implant interface within a vertebra that is loaded via a rod system. The focus is therefore on the anchoring in the bone and not on the fusion of two vertebrae. As a result, no conclusions can be drawn as to how it would behave in the event of vertebral fusion.
Comments
What direction were the forces repeated to pull out tested. Were only uniplanar considerations made. Were coupled motions evaluated?
Response No pullout was performed, a cranial caudal load was applied to the screw and the inclination of the screw relative to the bone was measured using an optical 3D measuring system
Reviewer 2 Report
Comments and Suggestions for Authors
Summary: "Does malpositioning of pedicle screws affect biomechanical stability in a novel quasistatic test setup?" is to investigate the biomechanical stability of correctly placed and misplaced pedicle screws using a cylcing craniocaudal stability.
Overall, the manuscript is writtent well and describes the testing methods and results. There are some recommendations below to improve the manuscript with some references.
Ethical statements are listed in the manuscript.
Line 166: Is the test using paired or unpaired data for their analysis? Please clarify.
Figure 5: Please correct the labels on the figure showing "k" and "F" on the right hand side and a Title stating "correct (c) and misplaced (F) screws".
Line 243-245: Please add 2-3 sentences to discuss some methods for trying to reduce screw pullout. See reference below.
Costăchescu B, Niculescu AG, Grumezescu AM, Teleanu DM. Screw Osteointegration-Increasing Biomechanical Resistance to Pull-Out Effect. Materials (Basel). 2023 Aug 11;16(16):5582. doi: 10.3390/ma16165582. PMID: 37629873; PMCID: PMC10456840.
Line 225-228: How do the failures of the screw pedicles affect the vertebrae? Please read reference below to help.
Loenen A.C., Noriega D.C., Wills C.R., Noailly J., Nunley P.D., Kirchner R., Ito K., van Rietbergen B. Misaligned spinal rods can induce high internal forces consistent with those observed to cause screw pullout and disc degeneration. Spine J. 2020;21:528–537. doi: 10.1016/j.spinee.2020.09.010.
Line 238 - 246: To improve the discussion, others investigated pedicle screws by fixation methods such as using finite element analysis? Does the author's force-inclination results show similarities to other's FEA studies performed that lead to high or low stresses on the vertebrae. Do the author's think their dynamic loading possibly alter these stress distributions to help explain their results? A reference example is below.
Xu M., Yang J., Lieberman I., Haddas R. Stress distribution in vertebral bone and pedicle screw and screw–bone load transfers among various fixation methods for lumbar spine surgical alignment: A finite element study. Med. Eng. Phys. 2018;63:26–32. doi: 10.1016/j.medengphy.2018.10.003.
Line 245-247: Does reference 24 explain why the malpositioning is a complication for children? Maybe explain in 2 sentences as why there maybe more difficulty with children such as they are "growing" patients unlike adults.
Comments on the Quality of English LanguageThe English is written well. No comments.
Author Response
Comments
Line 166: Is the test using paired or unpaired data for their analysis? Please clarify.
Response: We used a paired Wilcoxon test and included the information accordingly
Comments
Figure 5: Please correct the labels on the figure showing "k" and "F" on the right hand side and a Title stating "correct (c) and misplaced (F) screws".
Response: Thanks for this note. We corrected the labels.
Comments
Line 243-245: Please add 2-3 sentences to discuss some methods for trying to reduce screw pullout. See reference below.
Costăchescu B, Niculescu AG, Grumezescu AM, Teleanu DM. Screw Osteointegration-Increasing Biomechanical Resistance to Pull-Out Effect. Materials (Basel). 2023 Aug 11;16(16):5582. doi: 10.3390/ma16165582. PMID: 37629873; PMCID: PMC10456840.
Response: Thank you very much. Clarification on the point regarding the FEA studies.
Comments
Line 225-228: How do the failures of the screw pedicles affect the vertebrae? Please read reference below to help.
Loenen A.C., Noriega D.C., Wills C.R., Noailly J., Nunley P.D., Kirchner R., Ito K., van Rietbergen B. Misaligned spinal rods can induce high internal forces consistent with those observed to cause screw pullout and disc degeneration. Spine J. 2020;21:528–537. doi: 10.1016/j.spinee.2020.09.010.
Response Thank you for the comment. The aim of our study was to investigate malalignment with a cranial caudal load. The setup was based on ASTM 1717, but due to the force applied, it is mainly the interface between the bone and the screws that is loaded and the rod mentioned in the study plays a subordinate role.
Comments
Line 238 - 246: To improve the discussion, others investigated pedicle screws by fixation methods such as using finite element analysis? Does the author's force-inclination results show similarities to other's FEA studies performed that lead to high or low stresses on the vertebrae. Do the author's think their dynamic loading possibly alter these stress distributions to help explain their results? A reference example is below.
Xu M., Yang J., Lieberman I., Haddas R. Stress distribution in vertebral bone and pedicle screw and screw–bone load transfers among various fixation methods for lumbar spine surgical alignment: A finite element study. Med. Eng. Phys. 2018;63:26–32. doi: 10.1016/j.medengphy.2018.10.003.
Response: Thank you for the comment. We have just found another FE study relating to carinal-caudal movement initiation that supports the thesis that dynamic movement is more suitable than pullout. We have added this at the following point(line 239 to line 242). We were unable to incorporate these into our work. At the same time, we are also addressing your comment regarding the reduction of pullout
Comments
Line 245-247: Does reference 24 explain why the malpositioning is a complication for children? Maybe explain in 2 sentences as why there maybe more difficulty with children such as they are "growing" patients unlike adults.
Response: Thank you for the comment. The source does not provide more detailed information on the reasons for the revision, so no further statements can be made here.
Reviewer 3 Report
Comments and Suggestions for Authors
This article is a very important study. When the literature is reviewed, there are many studies on malposition, but there are very few studies on the biomechanics of malposition. Therefore, I think that this study should be accepted for publication.
Author Response
Thank you for the Review
Round 2
Reviewer 1 Report
Comments and Suggestions for Authors
Thank you for addressing my concerns in the current revision.
Reviewer 2 Report
Comments and Suggestions for Authors
The corrections are acceptable for publication. This is an interesting article.